



# A conservation palaeobiological approach to assess faunal response of threatened biota under natural and anthropogenic environmental change

Sabrina van de Velde[*1], Elisabeth L. Jorissen[*2], Thomas A. Neubauer[1,3], Silviu Radan[4], Ana Bianca Pavel[4], Marius Stoica[5], Christiaan G. C. Van Baak[6], Alberto Martínez Gándara[7], Luis Popa[7], Henko de Stigter[8,2], Hemmo A. Abels[9], Wout Krijgsman[2], Frank P. Wesselingh[1]

*S. van de Velde and E.L. Jorissen should be considered joint first author

[1]Naturalis Biodiversity Center, P.O. Box 9517, 2300 RA Leiden, the Netherlands. +31-71-7519264
[2]Palaeomagnetic Laboratory 'Fort Hoofddijk', Faculty of Geosciences, Utrecht University, Budapestlaan 17, 3584 CD Utrecht, the Netherlands
[3]Department of Animal Ecology & Systematics, Justus Liebig University, Heinrich-Buff-Ring 26-32 IFZ, 35392 Giessen, Germany
[4]National Institute of Marine Geology and Geoecology (GeoEcoMar), 23-25 Dimitrie Onciul St., 024053 Bucharest, Romania
[5]Department of Geology, Faculty of Geology and Geophysics, University of Bucharest, Bălcescu Bd. 1, 010041 Bucharest, Romania
[6]CASP, West Building, Madingley Rise, Madingley Road, CB3 0UD, Cambridge, United Kingdom
[7]Grigore Antipa National Museum of Natural History, Sos. Kiseleff Nr. 1, 011341 Bucharest, Romania
[8]NIOZ Royal Netherlands Institute for Sea Research, Department of Ocean Systems, 1790 AB Den Burg, the Netherlands
[9]Department of Geosciences and Engineering, Delft University of Technology, Stevinweg 1, 2628 CN Delft, The Netherlands

*Correspondence to:* Sabrina van de Velde (sabrina.vandevelde@naturalis.nl); Liesbeth L. Jorissen (e.l.jorissen@uu.nl)

**Abstract** Palaeoecological records are required to test ecological hypotheses necessary for conservation strategies as short-term observations can be insufficiently to capture natural variability and identify drivers of biotic change. Here, we demonstrate the importance of an integrated conservation palaeobiology approach to make validated decisions for conservation and mitigating action. Our model system is the Razim-Sinoie Lake complex (RSL) in the Danube Delta (Black Sea coast, Romania), a dynamic coastal lake system hosting unique Pontocaspian mollusc species that are now severely under threat. The Pontocaspians refer to an endemic species group that evolved in the Black Sea and Caspian Sea basins under reduced salinity settings over the past few million years. The natural, pre-industrial RSL contained a salinity gradient from fresh to mesohaline (18 ppm), until human interventions reduced the inflow of mesohaline Black Sea water into the lake system. We reconstruct the evolution of the RSL over the past 2000 years from integrated sedimentary facies and faunal analyses based on 11 age-dated sediment cores and investigate the response of mollusc species and communities to those past environmental changes. Three species associations ('marine', 'Pontocaspian', 'freshwater') exist and their spatiotemporal shifts through the system are documented. Variable salinity gradients developed, with marine settings (and faunas) dominating in the southern part of the system and freshwater conditions (and faunas) in the northern and western parts. Pontocaspian species have mostly occurred in the centre of the RSL within the marine–freshwater salinity gradient. Today, freshwater species dominate





the entire system, and only a single Pontocaspian species (*Monodacna colorata*) is found alive. We show that the human-induced reduced marine influence in the system has been a major driver of the decline of the endemic Pontocaspian biota. It urges for improved conservation actions by re-establishing a salinity gradient in the lake system to preserve these unique species.


Key words: *biodiversity crisis, mollusc assemblages, sedimentary facies, Pontocaspian, Razim-Sinoie Lake complex, Danube Delta, conservation strategy, salinity gradient*

## 1 Introduction

The emerging field of Conservation Palaeobiology aims to evaluate palaeobiological records to make informed
contributions to biodiversity conservation. A spectrum of Conservation Palaeobiology studies exists covering different time scales as well as biota (Birks, 2012; Dietl and Flessa, 2011; Kosnik and Kowalewski, 2016; Vegas-Vilarrubia et al., 2011). These provide important insights into resilience of species and communities under environmental change. Yet, few studies provide long-term observations, quantitative and environmental data, as well as direct guidance to conservation of threatened habitats and communities today (Cramer et al., 2017). A relevant
Conservation Palaeobiology model system should ideally include species and communities that are currently under threat together with a high spatiotemporal resolution geological record of past habitat dynamics and should contain both natural variation and anthropogenic influence. Altogether the data should result in policy relevant conservation proposals (Dietl and Flessa, 2011), which is often lacking in Conservation Palaeobiology studies (Albano et al., 2016; Helama et al., 2007; Kosnik and Kowalewski, 2016; Martinelli et al., 2017; Vegas-Vilarrubia et al., 2011).

Here, we report a unique Conservation Palaeobiology model system that combines a detailed historical record of environmental and faunal change with relevant proposals for conservation. The Razim-Sinoie Lake complex (RSL) consists of a set of coastal lakes/lagoons located at the southern Danube Delta along the Black Sea in Romania (Fig. 1). The system has a dynamic Late Holocene history of lake and barrier formation and of connectivity with the Danube River as well as the Black Sea that affected salinity gradients (Gâştescu, 2009; Giosan et al., 2006; Panin et
al., 2016; Panin and Jipa, 2002; Romanescu, 2013; Vespremeanu-Stroe et al., 2017). Since ancient Greek times, shipping ports have been present in the RSL (Bony et al., 2015; Breţcan et al., 2008; Romanescu, 2013), and from 1800 onwards, active human modification of the river and marine outlets shaped the system towards its present state (Breţcan et al., 2008, 2009). The RSL is known as prime habitat for currently threatened Pontocaspian species (Grossu, 1973; Popa et al., 2009, 2010). Within the area of the Danube Delta, a strong reduction of Pontocaspian
species occurrences has been documented for the past decades (Alexandrov et al., 2004; Popa et al., 2009). Before 1956, about 70% of the species living in the benthic zone had a Pontocaspian origin, and remaining species belonged to other brackish or freshwater species (Teodorescu-Leonte et al., 1956). After 2000, this relationship shifted completely and the dominant forms seem to be freshwater species (Catianis et al., 2018).

We aim to build a palaeobiological record from sediment cores in the RSL that shows faunal responses to past
natural and human-induced environmental variations. Insight into faunal development and resilience has direct relevance for outlining conservation strategies for Pontocaspian biota. The RSL is uniquely equipped for this study





as it is one of the largest areas where threatened Pontocaspian biota still occur in the Black Sea Basin and contains a detailed Late Holocene geological record of faunal and environmental change.

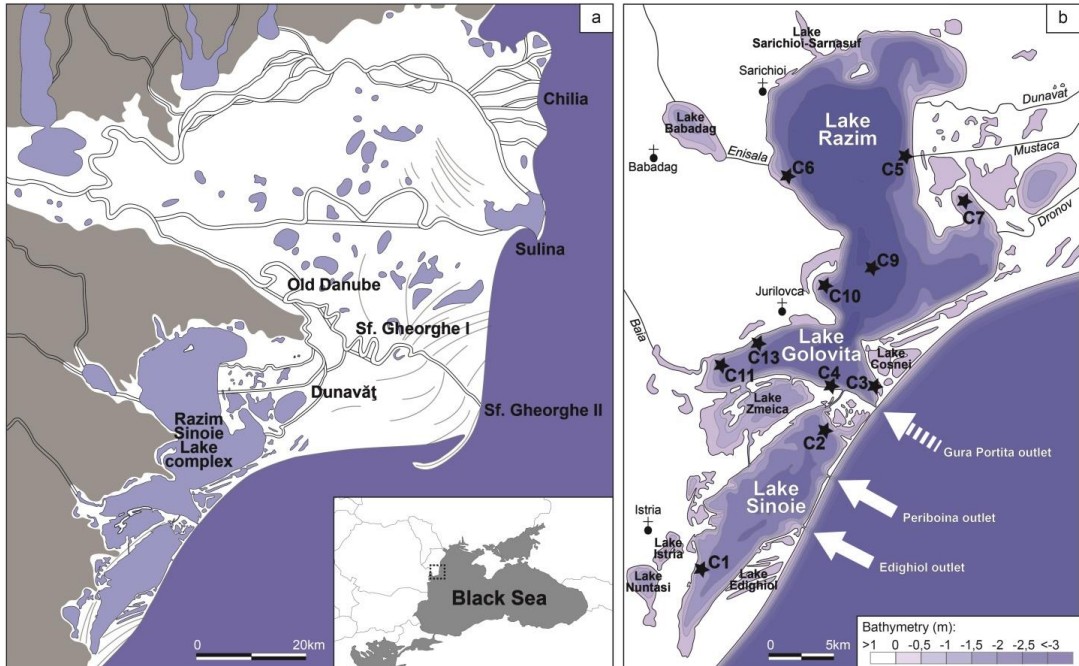

**Fig. 1: Location of study area along the Romanian Black Sea coast with core locations indicated by black stars. (a) Danube Delta and RSL (modified after Vespremeanu-Stroe et al., 2017). (b) RSL bathymetry with location of study cores (modified after Dimitriu et al., 2008). Two current marine outlets are indicated by white arrows; a third outlet (Gura Portita) was closed in the 1970ties and is indicated by a dashed white arrow.**

## 2 Regional setting

Over the past millennia, the RSL has evolved from a restricted marine embayment south of the Danube Delta into the highly restricted lake system of today (Vespremeanu-Stroe et al., 2017) (Fig. 1a). Much of the restriction has been attributed to prograding beach-barriers that were fed by sediments brought by long-shore currents from eroding Danubian deltaic lobes to the north (Dan et al., 2009; Giosan et al., 2006; Panin, 1989, 1997; Panin and Jipa, 2002; Ştefănescu, 1981; Vespremeanu-Stroe et al., 2013). Today, the RSL contains several large and many small shallow lakes of maximum 3.5 m deep and is supplied with water and sediments from the Dunavăţ Branch (Rădan et al., 1999).

Small-scale human modifications have occurred in the system since the ancient Greek dredged and modified access from the Black Sea to the ports of Istros/Histria and Orgame/Argamum (Bony et al., 2015; Breţcan et al., 2008; Romanescu, 2013). Major anthropogenic modifications impacted the system in the 20[th] century to increase freshwater aquaculture production. The digging of two channels between the RSL and the southern Danube branch during the first three decades of the 20[th] century and seven additional channels around 1950 (Breţcan et al., 2009;





Romanescu and Cojocaru, 2010) increased river influence in the RSL. At the same time, access to the Black Sea became restricted by the closure of the Gura Portiţa outlet around 1960–1970 (Breţcan et al., 2009; Breţcan and Tâmpu, 2008). Today, the Periboina and Edighiol channels remain the only (and highly restricted) connections between Lake Sinoie and the marine realm. As a result, the RSL has considerably freshened over the past 100 years (Catianis et al., 2018; Stănică, 2011).

Prior to major anthropogenic modifications, the RSL contained a salinity gradient between the northern–central parts dominated by fresh water and the southern part where elevated salinities occurred (Breţcan et al., 2009). The northern–central part consists of the three main lakes Razim (or Razelm), Goloviţa and Zmeica, and several marginal lakes such as Agighiol, Babadag, Coşna and Leahova. Salinities in this part of the system were reported between 0.4 and 2.5 g/L (Vadineanu et al., 1997). The southern part comprises the main lake Sinoie (also known as Sinoe), and a number of shallow and smaller lakes such as Istria, Nuntaşi and Edighiol. Here, salinities ranged between 0.5 and 6.5 g/L in the late 20th century (Vadineanu et al., 1997). Over the past decade, the entire system has become almost entirely fresh water: GeoEcoMar expeditions in 2017 estimated 0 psu in the northern–central parts based on mollusc species salinity tolerances and measured 0.1–0.6 psu in the southern part (Table S1). Salinity may vary seasonally depending on freshwater input from the Danube and through variations in storm activity over the Black Sea, causing additional mesohaline water to enter the RSL through the marine outlets (Breţcan et al., 2008; Dinu et al., 2015).

The RSL has been reported as prime habitat area for Pontocaspian species (Grossu, 1973; Popa et al., 2009, 2010), including endemic molluscs, ostracods, dinoflagellates, fish and crustaceans. Pontocaspian biota are a unique group of aquatic species that evolved within the Caspian Sea and Black Sea basins, as well as surrounding rivers and lakes, over the past few million years (Krijgsman et al., 2019). The basins experienced strong base level changes and periods of isolation and connection from the open ocean, characterised by variable, anomalohaline ("brackish") conditions (Grigorovich et al., 2003; Kosarev and Yablonskaya, 1994; Kurbanov et al., 2014; Yanina, 2014). Ever since the Middle Pleistocene, during times of interglacial sea-level highstands (similar to present-day), marine conditions and faunas occupied the main Black Sea Basin and pushed Pontocaspian species into coastal habitats such as deltas, lagoons and estuaries. During glacial lowstands, the Black Sea became isolated from the open ocean, freshened, and Pontocaspian biota occupied the entire basin (Krijgsman et al., 2019). Today, Pontocaspian biota in the Black Sea Basin are restricted to relatively small areas in lagoons, estuaries and deltas with salinity gradients along the coasts of Romania, Ukraine and Russia (including the Sea of Azov; Anistratenko, Khaliman, & Anistratenko, 2011; Mordukhai-Boltovskoi, 1979). Of these restricted areas, the RSL offers the largest area of Pontocaspian habitat. Pontocaspian species are currently under threat through habitat modification, pollution, poaching and invasive species (Popa et al., 2009; Shiganova, 2010; Zarbaliyeva et al., 2016).

## 3 Material and methods

We performed facies and fauna analyses on eleven shallow sediment cores. The cores of 0.5 to 3 m long were taken at 1 to 3 m water depth during two expeditions in October 2015 and July 2016 (Fig. 1b; Table 1). The cores were sampled with PVC pipes throughout the RSL and cut lengthwise in half back in the laboratory. Analyses were performed on one half; the other half was archived in a cold room at Utrecht University.





An age model was constructed using a variety of dating methods, including $^{14}$C measurements, analysis of downcore
distribution of $^{210}$Pb, palaeomagnetic secular variations and a known first occurrence date of an invasive species.
Five samples were dated with $^{14}$C measurements in the Center for Isotope Research at Groningen University. Dating
results were calibrated based on a time range with a 95% probability obtained from two reservoir ages. For the Late
Holocene, reservoir ages have been estimated at approximately 450 to 550 years in the Danube Delta (Bonsall et al.,
2004) and between 250 and 500 years in the Black Sea (Kwiecien et al., 2008).

The downcore distribution of $^{210}$Pb was assessed on four cores in the Ocean Systems Department of NIOZ, Texel.
Isotopic decay of $^{210}$Pb (half-life 22.3 years) was measured via its granddaughter radionuclide $^{210}$Po. We applied a
sampling resolution of 6 cm in the upper 14 cm and 12 cm in between 15 and 122 cm of the cores. Measurements
were performed on wet-sieved <63 µm sediment fraction. Sediments were spiked with 1000 µl of a standard solution
of $^{209}$Po and leached for 6 h at 105°C in a 10 ml solution of concentrated hydrochloric acid. After diluting with 40 ml

of demineralized water and 5 ml of ascorbic acid, silver platelets were immersed and left in the solution for 15 hours
at 80°C in order to collect the natural $^{210}$Po and the added $^{209}$Po by spontaneous electrochemical deposition. The Po
radionuclides were then counted by a Canberra alpha detector. The sediment depths where excess $^{210}$Pb had decayed
to below detection level were used as age tie point of 150 years BP.

We performed palaeomagnetic analyses on core C3 at the Palaeomagnetic Laboratory Fort Hoofddijk at Utrecht

University. Oriented 0.5 cm$^3$ plastic cubes were pushed in the sediment core in two overlapping rows giving an
average sampling resolution of 0.5 cm. The sediments in these cubes were then stepwise demagnetized using
alternating fields (AF) up to a maximum of 100 mT on a robotized sample handler controller attached to a horizontal
2G Enterprises DC SQUID cryogenic magnetometer (Mullender et al., 2016). Changes in the magnetisation direction
were recorded along the core and compared with regional archaeomagnetic data (Kovacheva et al., 1998, 2014).

Finally, the first arrival of the New Zealand invasive snail *Potamopyrgus antipodarum* in Europe in 1859 (Ponder,
1988) provided an additional maximum age tie point. We calculated sedimentation rates for all cores with dates
available and extended the results to the remaining cores through correlation of sedimentary facies.

Sedimentary facies were characterized using lithological and sedimentological criteria and used to develop a facies
model. Cores were digitally photographed and logged at a millimetre-scale. Sediment colours were defined using the

Munsell Soil Colour Charts. Sedimentary features such as grain size variations, laminations and cross-stratification
were described. Various types of inclusions were documented, such as the presence of shells, organic material as
well as trace fossils. Detailed sedimentological observations permitted the understanding of variations in depositional
energy regimes and sedimentary processes and helped tracking palaeoenvironments through time.

Mollusc samples, consisting of a block of sediment of 1 cm thickness and 5 cm diameter, were taken at 6 cm

intervals. Samples were washed and sieved over a 0.5 mm sieve. Samples rich in fauna (estimated >300 specimens)
were split to facilitate counting. Molluscs were identified to species level and the amount of individuals for each
species was counted. For bivalves, one valve was counted as half an individual, whereas for gastropods, a fragment
with a protoconch was considered one individual. Final counts were rounded up to the next integer. Species were
classified into groups with different salinity requirements corresponding to their different evolutionary origins (i.e.

freshwater, Pontocaspian and marine), to be able to follow the threatened Pontocaspian species through time and
space (Table S2).



For the current species distribution within the RSL, we used qualitative data of three expeditions carried out since 2015 (Table S1). Van Veen grab and dredge samples from 77 stations obtained during a 2017 survey of GeoEcoMar were investigated for living molluscs. Together with the data from two surveys in 2015 and 2016 by members of the
team of authors specifically in search for living Pontocaspian molluscs in the RSL, a distribution map of species was made.

We used three taphonomic criteria on the mollusc faunas to score for the fidelity of samples, i.e. fragmentation, dissolution and abrasion (details in Table S3). These characters yield information about transport, energy and post-burial processes like compaction and dissolution and may point to reworking/time-averaging of a sample (Erthal et
al., 2011; Kidwell, 2013). Samples with high taphonomic alteration (any of the three taphonomic scores higher than 3; Table S3) were excluded from statistical analyses as they are likely to represent time-averaged/mixed assemblages.

For palaeosalinity estimates, we used published autecological tolerances of species (Table S2). However, published upper salinity tolerances are often laboratory-based or derive from the unique Baltic Sea (Zhadin, 1952) and are not
very informative as to the actual preferences of species in the RSL. We therefore assigned species to their optimum salinity habitat within the system: freshwater, lower/upper oligohaline and lower/upper mesohaline (sensu Strydom, Whitfield and Wooldridge, 2003, Table S2). The Black Sea has a salinity of 18 psu, which is used as the maximum salinity in our classification, although the salinity in the southern portion of the RSL is occasionally higher due to extensive summer evaporation (Dinu et al., 2015).

We performed a variety of multivariate analyses and statistical tests to explore the spatiotemporal species distribution and associations and test for potential associations with environmental variables. Samples with fewer than thirty specimens were excluded from the analyses to avoid the effect of small sample size on species richness.

The similarity among species compositions of individual samples was explored by means of a non-metric multidimensional scaling (nMDS). Data were square rooted and subjected to Wisconsin double standardization,
where species are first standardized by maxima and then samples by sample totals. The nMDS was computed from a Bray-Curtis dissimilarity matrix. Salinity (i.e. weighted average of species' optimum salinity per sample) and grain size (using the upper bound of each category following the Wentworth scale) were fitted as 2D surfaces to the nMDS ordination plot to test if differences in species compositions are related to these parameters (compare Hauffe et al., 2011; Neubauer, Harzhauser, Mandic, Kroh, & Georgopoulou, 2016). The applied surface fitting uses restricted
maximum likelihood estimation and an isotropic smoother employing thin-plate regression splines. For visualization purposes, the nMDS plot was rotated so that the first axis is parallel to the salinity gradient and the second to grain size.

K-means partitioning was used to determine if the three evolutionary species groups correspond to ecological communities (Legendre, 2005). The data were Hellinger-transformed, i.e. square-rooting the relative abundances of
count data, and standardized (Borcard et al., 2011; Legendre and Gallagher, 2001). This procedure has been recommended for clustering of species abundance data because it limits the influence of high abundance values (Legendre and Gallagher, 2001; Rao, 1995). We used the twenty most abundant species to avoid a bias from rare species (Borcard et al., 2011). We chose $k$ to range between 2 and 10 in order to find appropriate groups representing different environmental types. To find significant species associations, we calculated Kendall's $W$ coefficient of



concordance for each of the partitioned datasets (Legendre, 2005). This method tests for the most encompassing assemblages, hence the smallest number of clusters with the largest number of positively and significantly associated species (Borcard et al., 2011). For each $k$, a global Kendall's $W$ test was run to estimate if the identified species groups were significantly associated. An *a posteriori* test was conducted to evaluate the contribution of each species to the groups (Borcard et al., 2011; Legendre, 2005). Species groups that were not globally significant, as well as

species that were negatively or not significantly associated, were excluded. For the definition of species associations, we chose the solution with $k$ being smallest and yielding groupings that reflect ecologically consistent units.

Finally, we tested for differences of the relative abundances of each of the above defined ecological associations per sample across facies type. A Shapiro-Wilk test was conducted to test for normality of the relative abundances per association. Since normality was rejected in all cases, we performed for each association a Kruskal-Wallis rank sum

test to assess whether the medians of relative abundances differ globally across facies types. In addition, pairwise Wilcoxon rank sum tests were computed to test for differences of median relative abundances between individual facies.

All analyses were performed in R v. 3.3.3 (R Core Team 2017), using package 'vegan' v. 2.4-4 (Oksanen et al., 2015)

**4 Results**

**4.1 Age model**

We obtained 13 age tie points showing that the RSL record covers slightly over 2000 years (Fig. 2, Table S4). An average sedimentation rate of 0.113 cm/years across the entire lake system was deduced (Table S5). Sedimentation rates are estimated as 0.076 cm/years for the lower parts of the cores and as 0.144 cm/years for the upper parts.

Sedimentation rates in the RSL are high in comparison to other regional examples of Miocene–Pliocene deltaic and lacustrine environments (Jorissen et al., 2018; De Leeuw et al., 2013), which may be explained by a lack of post-depositional compaction of the sediments in these young lagoon and lacustrine sediments.














**Fig. 2: Overview of core data. From left to right each core: core photograph, lithology, facies, fauna relative abundance per species group based on origin/evolution and estimated palaeosalinities.**

**4.2 Facies model**

We identified six facies that represent specific sedimentary processes and depositional environments (Fig. 2; Table S6). Facies F1 consists of medium-grey (GLEY 2-4/5B) silts to coarse-grained sands with many reworked millimetre-scale shell fragments. We interpret this facies as deposited under high energies in lagoon environments exposed to open marine conditions. Facies F2 is made of medium-grey (GLEY 2-5/5B) shell-rich silty clays to coarse-grained sands, with many millimetre- to centimetre-scale reworked shell fragments and complete shells. This

facies is interpreted as sediments washing over sand barriers during storms or after human interventions and deposited in lagoon environments. Facies 3 consists of greenish-grey (GLEY 1-5/10Y) to olive grey (5Y-5/2) silts or clays with some millimetre-scale horizontal laminations. We interpret this facies to be deposited under fluctuating energy regimes, as a combination of suspension deposits and flood events within shallow lagoon to lacustrine environments. Facies 4 is made of olive grey (5Y-5/2) organic-rich silts and clays. Many millimetre- to centimetre-

scale terrestrial organic material fragments, including many reeds incorporated vertically or horizontally into the sediments. This facies is considered to represent low depositional energies in shallow lacustrine environments. Facies 5 consists of dark olive grey (5Y-3/2) to black (5Y-2.5/2) organic-rich clays. These sediments contain abundant millimetre- to centimetre-scale terrestrial organic fragments, forming peat layers. We interpret this facies to represent very low energy stagnant water bodies in swamp areas. Finally, facies 6 is made of olive-brown (2.5Y-4/3)

to brown (10YR-4/3) clays without structures. We interpret this facies to be deposited under flocculation of clay particles in very low energy environments, affected by sediment oxidation. These deposits represent the most recent sediments deposited in the lacustrine system (the so-called active layer). The six facies highlight various depositional energies in the system. Dynamic sedimentary processes are dominant in lagoon environments with preferential deposition of shelly sands, whereas more stable sedimentary processes are prevalent in lacustrine environments with

preferential deposition of organic-rich clays.

**4.3 Mollusc composition and palaeosalinity**

A total of 235 samples yielded 16,587 mollusc specimens belonging to 44 species (Table S7).

Most species are typical for freshwater environments (24 species), 12 species are of marine origin, and eight species

have a Pontocaspian character (Table S2). Salinity tolerances of the latter group typically include a range from freshwater to oligohaline settings. The three species groups are present in all cores, showing variable presence through time and space (Fig. 2). The freshwater group is dominated by *Dreissena polymorpha*, *Valvata piscinalis* and *Gyraulus crista*. Marine species are mainly opportunistic species that live in mesohaline conditions and include *Lentidium mediterraneum*, *Ecrobia maritima* and *Abra segmentum*. The Pontocaspian group consists of *Adacna*

*fragilis, Clathrocaspia knipowitschii, Dreissena bugensis, Hypanis plicata, Monodacna colorata* and *Theodoxus* spp.





Most often, individual samples include a mix of the three salinity groups, while some show a 100% domination of fresh or marine species. Pontocaspian species rarely dominate a whole sample (only twice: C10 at 83 cm and C13 at 42 cm).

The distribution data of extant species show an almost complete domination by freshwater species in the entire RSL

(Fig. 3, Table S1). In Lake Razim, Lake Goloviţa, Lake Zmeica, as well as along the Jurilovca Canal, the main species found alive are freshwater species. The only Pontocaspian species found in the RSL system is *Monodacna colorata,* and in very low numbers (1–5 individuals per station). In Lake Sinoie, very few living species were found in 2 out of 15 stations, all of which were freshwater taxa. The two field trips in 2015 and 2016 that specifically focused on finding living Pontocaspian Lymnocardiinae resulted in a few additional locations for living *Monodacna*

*colorata* (Fig. 3). During our expeditions in the RSL in search for living molluscs, no other living Pontocaspian species besides *Monodacna colorata* were found, nor any marine species.

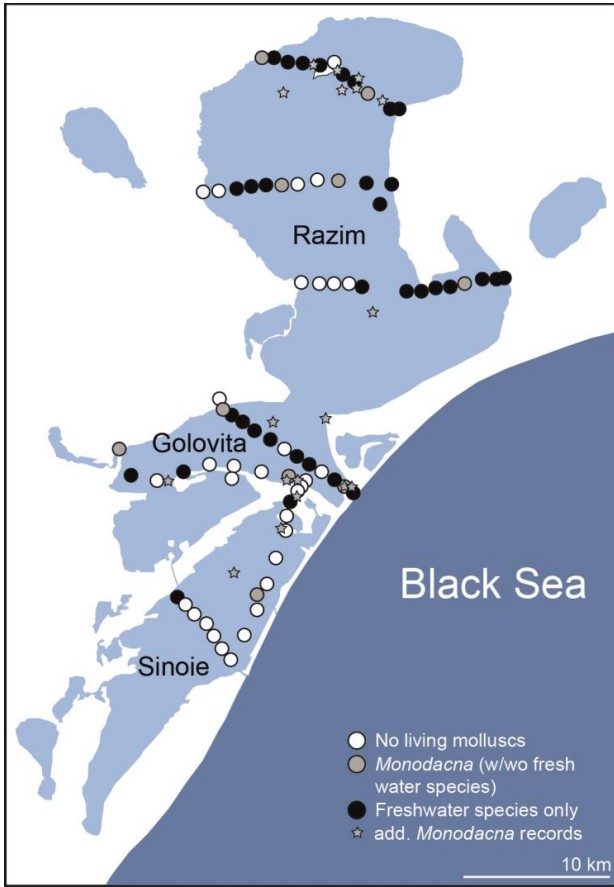

**Fig. 3: Sampling locations of three expeditions in 2015–2017. The first (September–October 2015) and second expedition (July 2016) were in particular search of living Pontocaspian molluscs. The third expedition consisted of six transects**
**sampled by GeoEcoMar in 2017 in search of any living mollusc.**

Our age-dated environment model shows that the salinity gradients have dynamically shifted through the system over time (Fig. 4). The southern part (cores C01-C02-C03-C04) has maintained mesohaline conditions almost during the





entire time interval. The northern part (C05-C06-C07) has been freshwater to oligohaline with intervals of increased salinities. Core 7 contains unusual high salinity associations between 500 AD and 1900 AD compared to the

surrounding cores. The cores taken in the centre of the system (C09-C10-C11-C13) show several oscillations between freshwater and mesohaline conditions. The majority of the cores indicate mesohaline conditions at the base.

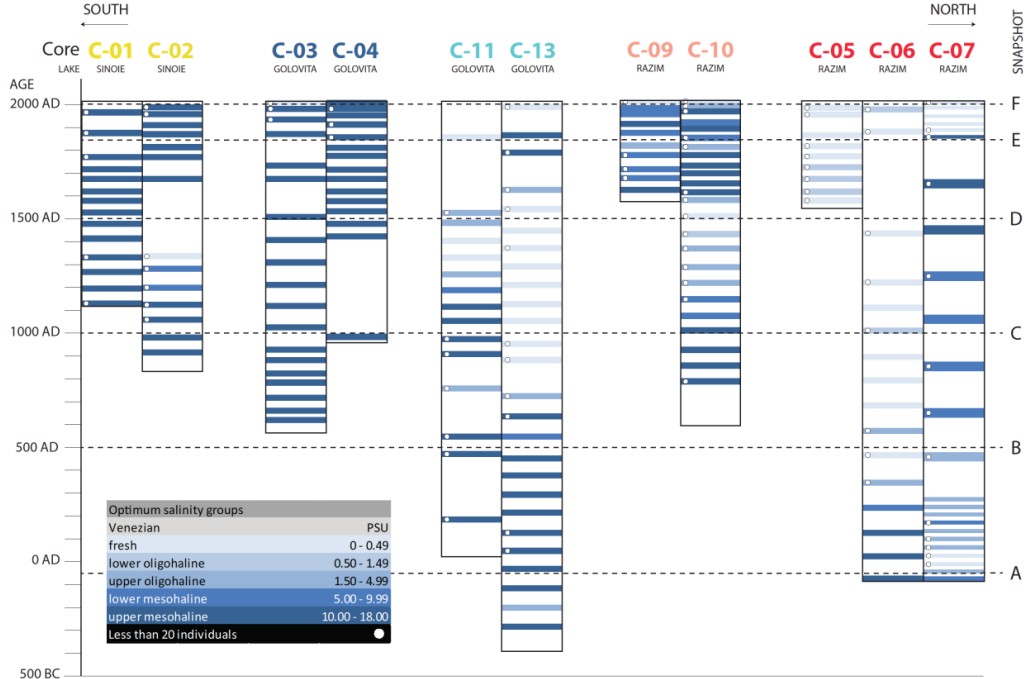

**Fig. 4: Spatiotemporal salinity variations in the RSL. For each sample, the salinity was calculated by weighted averaging of the species' optimum salinities. Salinity categories adapted from Strydom, Whitfield and Wooldridge (2003). Snapshots**
**referred to in the Discussion are indicated with letters A-F.**

### 4.4 Statistical analyses

The nMDS shows that in the 2000-year record most freshwater species lived in the northern part of the system (Razim), while the centre and south (Goloviţa and Sinoie) were dominated by species that tolerate higher salinities (nMDS stress = 0.173) (Fig. 5, Table S8). More precisely, the southern lakes have been almost permanently occupied

by mesohaline species, and freshwater species dominated the northern part of Lake Razim and western Lake Goloviţa most of the time. Pontocaspian species occurred in various parts of the system through time, yet they were almost always present in the central–northern parts and almost entirely lacking in the southern part. The surface fitting revealed a strong association between species composition and salinity ($R^2_{adj}$ = 0.945, deviance explained = 95.1%, P < 0.001) and, to a lesser extent, grain size ($R^2_{adj}$ = 0.533, deviance explained = 57.6%, P < 0.001),

respectively (Table S8).





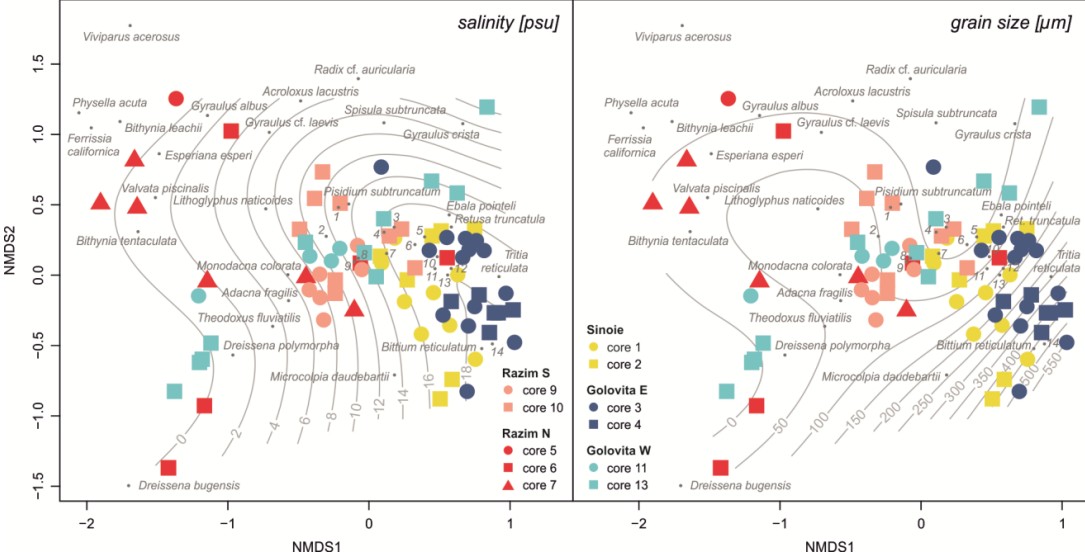

**Fig. 5: NMDS ordination plot of species compositions across samples grouped into lake regions (stress = 0.173). Optimum salinity and grain size were fitted as two-dimensional smooth surfaces to illustrate the associations with species composition. Species marked with numbers: 1 – *Planorbis planorbis*, 2 – *Clathrocaspia knipowitschii*, 3 – *Potamopyrgus antipodarum*, 4 – *Theodoxus danubialis*, 5 – *Planorbarius corneus*, 6 – *Abra segmentum*, 7 – *Rissoa membranacea*, 8 – *Valvata macrostoma*, 9 – *Hypanis plicata*, 10 – *Mytilaster minimus*, 11 – *Ecrobia maritima*, 12 – *Parthenina interstincta*, 13 – *Cerastoderma glaucum*, 14 – *Lentidium mediterraneum*.**

The K-means partitioning for two, three, four and five groups yields significant results for Kendall's *W* coefficient of concordance; higher *k* produced groups partly consisting of single species, which do not allow testing for concordant groups. Only for *k* = 2, all groups were globally concordant; for each of *k* = 3 through to *k* = 5, one group was not concordant (Table S8). The *a posteriori* test for *k* = 2 generally distinguished between a marine group (but including *Potamopyrgus antipodarum* and *Theodoxus danubialis*) and a freshwater–Pontocaspian group (Table S8). The two concordant groups identified for *k* = 3 both consist of a mixed composition: one cluster was dominated by marine species, including again the freshwater species *P. antipodarum* and *T. danubialis*, the other one was composed of Pontocaspian species along with freshwater *Dreissena polymorpha* and marine *Rissoa membranacea*. The *a posteriori* test for *k* = 4 identified three groups that match the groups based on species origin, with the only exceptions that the marine *R. membranacea* clustered with the Pontocaspian group (Table 2). For *k* = 5, the Pontocaspian group was split up into bivalves (including the freshwater *D. polymorpha*) and a mixed gastropod group including *Clathrocaspia, Rissoa* and *T. danubialis*. While the marine group was still clearly demarcated, hardly any of the freshwater species were significantly associated.



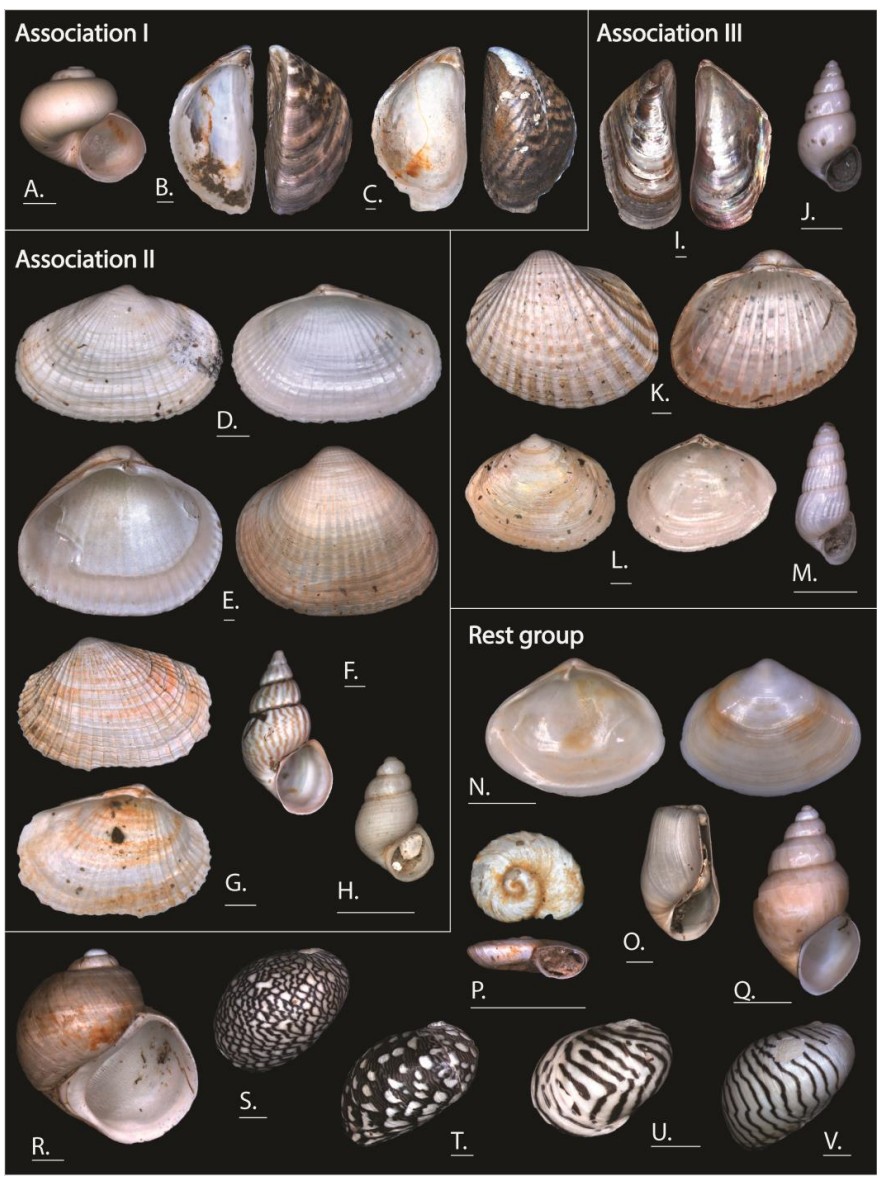

**Fig. 6: Overview of the twenty most abundant mollusc species grouped according to the results of Kendall's *W* coefficient of concordance (for *k* = 4). A.** *Valvata piscinalis* (RGM.1309841, Core C7, depth 6 cm). **B.** *Dreissena polymorpha* (RGM.1309827, C7 - 6 cm). **C.** *Dreissena bugensis* (RGM.1309846, C5 - 18 cm). **D.** *Adacna fragilis* (RGM.1309835, C2 - 18 cm). **E.** *Monodacna colorata* s.l. (RGM.1309823, C7 - 14 cm). **F.** *Rissoa membranacea* (RGM.1309830, C3 - 48 cm). **G.** *Hypanis plicata* (RGM.1309845, C9 - 3 cm). **H.** *Clathrocaspia knipowitschii* (RGM.1309843, C11 - 102 cm). **I.** *Mytilaster minimus* (RGM.1309838, C3 – 24 cm). **J.** *Ecrobia maritima* (RGM.1309831, C3 - 48 cm). **K.** *Cerastoderma glaucum* (RGM.1309844, C13 - 24 cm). **L.** *Abra segmentum* (RGM.1309821, C1 - 48 cm). **M.** *Parthenina interstincta* (RGM.1309832, C3 - 48 cm). **N.** *Lentidium mediterraneum* (RGM.1309837, C4 - 12 cm). **O.** *Retusa truncatula* (RGM.1309828, C2 - 42 cm). **P.** *Gyraulus crista* (RGM.1309840, C5 - 54 cm). **Q.** *Potamopyrgus antipodarum* (RGM.1309836, C2 - 18 cm). **R.** *Lithoglyphus naticoides* (RGM.1309842, C5 - 18 cm). **S.** *Theodoxus fluviatilis* (RGM.1309826, C11 - 66 cm). **T.** *T. fluviatilis* (RGM.1309824, C11 - 78 cm). **U.** *Theodoxus danubialis* (RGM.1309839, C3 - 24 cm). **V.** *T. danubialis* (RGM.1309834, C2 - 30 cm). Scale bars are 1 mm.




Of all the clustering levels tested for, $k = 4$ yielded the ecologically most consistent groupings (Fig. 6). Indicator species for Association I are *Dreissena polymorpha*, *D. bugensis* and *Valvata piscinalis*. The *Dreissena* species are filter feeding mussels that require hard/firm substrates such as shells or sticks (Gittenberger et al., 2004). *Valvata piscinalis* is a herbivore species that occurs on muddy bottoms in standing to slowly moving waters rich in

plants(Glöer, 2002). All three species are common freshwater species but are able to tolerate salinities up to 5 psu (Glöer, 2002).

Association II is characterized by mainly Pontocaspian species, i.e. *Adacna fragilis*, *Monodacna colorata*, *Hypanis plicata* and *Clathrocaspia knipowitschii*. In general, *Adacna* spp. occur on muddy, rarely sandy, bottoms. In the Caspian Sea, they can tolerate salinities between 4–14 psu (Bogutskaya et al., 2013). *Monodacna colorata* inhabits

muddy and sandy-muddy substrates and has its optimum habitat between 0.03–4 psu, but can tolerate higher salinities (Bogutskaya et al., 2013). *Hypanis plicata* is a filter feeder that prefers silty-sandy to clayey bottoms between 0.5 and 30 m (Bogutskaya et al., 2013). In the Caspian Sea, it prefers salinities between 4 and 8 psu (Bogutskaya et al., 2013), yet in the Black Sea, populations occur in lower salinities (Popa et al., 2009). They have also been observed in the freshwater Volga Delta (Yanina et al., 2010). *Clathrocaspia knipowitschii* is a herbivore

gastropod that has been found at depths of 1.5–3 m in deltaic areas with salinities of 0–1.5 psu (Boeters et al., 2015). In addition to the Pontocaspian species, the marine grazing gastropod *Rissoa membranacea* clusters with this association. This species prefers 15–19 psu but can tolerate salinities below 15 psu (Rehfeldt, 1968).

Association III is defined by marine species: *Mytilaster minimus*, *Ecrobia maritima*, *Cerastoderma glaucum*, *Abra segmentum* and *Parthenina interstincta*. *Abra*, *Cerastoderma* and *Ecrobia* are capable of withstanding extreme

variations in temperature and salinities between oligohaline and hypersaline (Gontikaki et al., 2003; Kevrekidis et al., 2009; Kevrekidis and Wilke, 2005). *Parthenina* and *Mytilaster* are mesohaline taxa that prefer salinities above 15 psu (Table S2).

The remaining seven of the twenty most abundant species are either assigned to a non-concordant group or are not significantly associated. This outgroup contains a mixture of ecological types, including strictly marine taxa like

*Lentidium mediterraneum* and *Retusa truncatula*, as well as typical freshwater species (*Lithoglyphus naticoides*, *Gyraulus crista*) and species that can tolerate a broad range of salinities (*Theodoxus danubialis*, *T. fluviatilis*, *Potamopyrgus antipodarum)*. These species are variably assigned to different clusters in other solutions of the K-means clustering (Table S8), which suggests that they might not be characteristic for a specific environment or belong to ecological groups that could not be identified by our analyses.

The relative abundances of the three associations are partly related to facies type (Fig. 7). Association I is dominant in the low-energy lacustrine facies F4 and F6, latter of which is typical of the modern setting. The marine Association III is most common in facies F3, characteristic for shallow lagoon to lacustrine environments, and F5, typical of stagnant swampy areas. In contrast, the Pontocaspian-dominated Association II is evenly distributed across all facies types. This result is also confirmed by the Kruskal-Wallis rank sum tests: Association I ($\chi^2 = 37.367$, P <

0.001) and Association III ($\chi^2 = 23.431$, P < 0.001) show significant differences of the median relative abundances across facies types, while Association II does not ($\chi^2 = 7.751$, P = 0.101). Pairwise Wilcoxon tests yielded six significant differences for Association I (F1-F2, F1-F4, F1-F6, F2-F3, F3-F4, F3-F6) and three for Association III (F1-F3, F3-F4, F3-F6) (Table S8).



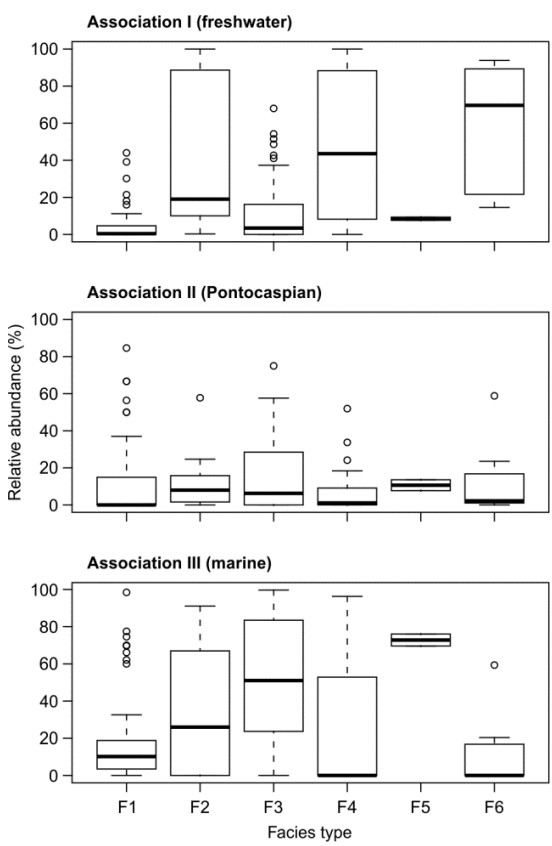

Fig. 7. Boxplots show the distribution of relative abundances of the associations across the six facies types.

## 5 Discussion

### 5.1 Evolution of the RSL

We subdivided the Late Holocene evolution of the RSL into six phases, represented by snapshots A–F (Fig. 4, Fig.
8). The phases are strongly related to the development of the Danube Delta (Giosan et al., 2006; Panin, 1989, 1997;
Panin et al., 2003; Ştefănescu, 1981; Vespremeanu-Stroe et al., 2017), as well as modern human modifications of the
RSL (Breţcan et al., 2009; Romanescu and Cojocaru, 2010).









**Fig. 8: Snapshot reconstructions of the evolution of the RSL and their mollusc biota. The names of major sand barriers are indicated in yellow, those of deltaic lobes in black. The names in parentheses and italic font are currently inactive lobes. Pie charts indicate the relative abundance of the three associations in the time interval of ± 50 years of the indicated snapshot: blue – Association I (freshwater); green – Association II (Pontocaspian), orange – Association III (marine), grey – rest group. Water colours indicate a salinity gradient: blue is Black Sea influence (18 psu); green is river influence (0 psu). Note the freshening of the system and according changes in species associations with the decreasing influence of mesohaline waters from the Black Sea.**

Around 50 BC, the RSL was a restricted embayment west of the Black Sea, formed by a barrier complex south of the Danube Delta (Fig. 8A). At that time, the Sf. Gheorghe I lobe was no longer active for the benefit of the southern Old Dunavăţ (D1) lobe (Vespremeanu-Stroe et al., 2017), equivalent to the old Coşna Delta described earlier (Panin, 1989, 1997). Erosion of the deltaic lobes by strong longshore currents (Dan et al., 2009) had shaped an asymmetric wave-dominated delta to the north (Bhattacharya and Giosan, 2003; Preoteasa et al., 2016). Sediment drifting had created the Zmeica, Istria and Pahane sand barriers (Bony et al., 2015; Giosan et al., 2006; Vespremeanu-Stroe et al., 2017). The analysed facies show the separation of a protected marine bay in the west and an exposed lagoon in the north (Fig. 2). Both the presence of marine Association III species in all cores mesohaline conditions (Fig. 4) confirm the idea of a marine bay and show the mesohaline waters from the Black Sea dominated most of the RSL. Association I (freshwater) and Association II (Pontocaspian) dominated in the northeast, close to the river inflow.

The lagoon underwent further isolation from the Black Sea by a second barrier complex at c. 500 AD (Fig. 8B). The Old Dunavăţ (D1) developed into the New Dunavăţ (D2) lobe (Vespremeanu-Stroe et al., 2017), corresponding to the so-called Sinoie Delta (Panin 1997, 1989). The Lupilor (Vespremeanu-Stroe et al., 2013) and the Saele (Bony et al., 2015) sand barriers were created by progradation of the deltaic lobe (Vespremeanu-Stroe et al., 2013). The input of freshwater from the Danube increased in the north as the RSL became increasingly isolated from the Black Sea, expressed by swamp settings in the northwest and exposed lagoon environments near outlets (Fig. 2). The relative abundance of Association III decreased, while Association I and II increased, especially in the north.

At around 1000 AD, a third barrier complex expanded the lagoon system southwards (Fig. 8C). Sediment progradation from the Sf. Gheorghe II lobe had created the Chituc sand barrier (Vespremeanu-Stroe et al., 2017), and further prolonged the Saele sand barrier (Bony et al., 2015). Together they shaped a larger lagoon system, forming the outlines of the modern lakes Razim, Goloviţa and Sinoie. The outlet became smaller and decreased the influence of the mesohaline Black Sea. Sediments deposited in the south of the system, near the main outlet, indicate exposed lagoon environments, whereas the central and northern parts of the system formed more restricted lagoon and lacustrine environments (Fig. 2). Association III (marine) shifted southwards, while Association I (freshwater) dominated near the river mouths. Association II (Pontocaspian) dominated the centre of the system.

This situation continued until the end of the Middle Ages (1500 AD), when the lagoon probably experienced strong coastal erosion causing the opening of a second outlet in the north-east (Fig. 8D). Sediment that had eroded in the north drifted southward and further prolonged the Chituc sand barrier (Vespremeanu-Stroe et al., 2017). Marine influence became larger in the northern part, while maintaining river inflow from the west. A threefold increase of the relative abundance of Association III and the almost complete disappearance of the Pontocaspian-dominated Association II indicate a strong turnover of species and a salinity change towards mesohaline (5–18 psu) conditions (Fig. 4). In the south, where the direct influence of the Black Sea was maintained, Association III remained



dominant. The western part of the system formed a more protected shallow lacustrine environment (Fig. 2), where freshwater Association I was dominant.

Around 1850 AD, the outline of the system became very similar to the current RSL (Fig. 8E). Sediment progradation had stopped and the coast underwent strong erosion (Dan et al., 2009). Sediments deposited in Lake Sinoie are typical of shallow lagoon environments (Fig. 2), but without any input from rivers. Association III remained dominant here. The northern part of the system was a protected shallow lacustrine environment (Fig. 2), where Association I gained in relative abundance following increased Danube inflow. Lake Goloviţa was connected to the

Black Sea via the Gura Portiţa outlet but also experienced riverine influence. In this exposed lagoon environment (Fig. 2), Pontocaspian species of Association II expanded. They had probably previously found refuge in small patches in the north on the boundary between mesohaline and freshwater.

In the past century (Fig. 8F), anthropogenic activities (closure of marine outlets, opening of channels connecting to the Danube) caused a salinity decrease of the RSL (Alexandrov et al., 2004; Breţcan et al., 2009; Romanescu and

Cojocaru, 2010). Human interventions started at the beginning of the 20[th] century with freshening the lagoon system for economic reasons (Alexandrov et al., 2004; Breţcan et al., 2009). Two channels were dredged at the beginning of the 20[th] century and seven additional channels around 1950 in order to increase the influence of the Danube River. Around 1970, the Gura Portiţa outlet was closed to limit marine connections with the Black Sea. In Lake Razim, mollusc associations I and II dominated, while the top samples from the cores indicate a decrease of Association III

in the south. Yet, a total freshening of the fauna, as seen in the current species occurrences (see below), has not yet been archived within the active layer.

During the past few years, the isolation of the RSL system led to the development of more restricted lacustrine environments with lowered salinities. Observed living mollusc occurrences show a fauna dominated by freshwater species (Fig. 3). A comparable study on fish populations in the RSL showed a similar freshening signal in 2002

(Alexandrov et al., 2004).

## 5.2 Optimum habitat and resilience of Pontocaspian species

The species groups defined by evolutionary origin almost entirely correspond to ecological associations, which in turn relate to environmental settings. Species distributions shifted through the RSL in response to changes in the environment. Increasing Black Sea influence matched with decreasing freshwater species occurrences. In turn, in

periods of freshening, strictly mesohaline species such as *Mytilaster minimus* and *Parthenina interstincta* disappeared.

Our analyses show various correlations between species associations and salinity and to a lesser extent grain size. In general, freshwater species are found in more muddy-clay environments, marine species in sandy environments, and Pontocaspians in the transition area between clay and sand. Similar correlations between species, salinity and

sediment have been demonstrated by other studies (Nanami et al., 2005; Teske and Wooldridge, 2003; Ysebaert and Herman, 2002).

Optimum Pontocaspian habitats were associated with the presence of large enough areas of freshwater to oligohaline conditions. Pontocaspian species rarely dominated the faunas and always co-occurred with either freshwater or




marine species. Pontocaspian species are in general well adapted to salinity fluctuations in the freshwater to
mesohaline domain (Krijgsman et al., 2019). Such conditions have existed throughout the Late Holocene in the RSL.
Since 2000, however, the mixed Pontocaspian assemblages gave way to freshwater assemblages (Catianis et al.,
2018).

Around 1500 AD, Pontocaspian species abundances dropped throughout the RSL. The western part became
dominated by freshwater species, and marine species dominated almost the entire remaining system. The increase in
salinity caused by the connection to the Black Sea via the second outlet might have happened suddenly, giving room
to marine species to expand and replace Pontocaspian species. Afterwards, the RSL freshened and Pontocaspian
species re-established in the central–northern parts.

Some apparent incompatibilities occur, including the presence of the marine *Rissoa membranacea* in the
Pontocaspian community. This incompatibility may have resulted from the mixing of successive communities by,
e.g. bioturbation in a single interval, even although we tried to avoid such influence by using taphonomic filters.

### 5.3 Human impact and current species distribution

Major human impact in the system occurred in the 20[th] century with the simultaneous increase of Danube input and
decrease of Black Sea connectivity leading to an overall freshening of the system. Channels dug around 1900 and
1950, as well as the closure of the main marine outlet around 1970, increased the fresh water inflow from the Danube
Delta and limited the marine inflow from the Black Sea into the RSL (Breţcan et al., 2008; Romanescu and
Cojocaru, 2010). While the optimum Pontocaspian habitats were usually contained in the central–northern parts of
the system, they have shifted throughout the whole system over the past two thousand years. However, the
freshening has continued until present-day, and currently the whole complex has an estimated salinity of between 0
in the north and 0.6 psu in the south (Table S1). This freshening has resulted in the dominance of freshwater species
across the entire RSL and the almost complete disappearance of Pontocaspian species. Similar trends in community
shifts in the RSL have been reported for Pontocaspian fish (Alexandrov et al., 2004).

The near complete break-down of the previous salinity gradient has the potential to permanently eradicate
Pontocaspian habitat in the RSL. Pontocaspian biota has been able to deal with many salinity changes in the past
(Krijgsman et al., 2019) but might not overcome the current situation in the RSL complex where refuge habitats may
be lacking. Previously, there was always a small salinity gradient available somewhere in the system where
Pontocaspian could find refuge, but if the entire system further freshens, the remaining Pontocaspian species may
face local extinction. We see a severe decline in Pontocaspian species numbers based on the observations of living
species over the past few years. The only Pontocaspian species found living is *Monodacna colorata*, which is
particularly suited to survive in lower salinity ranges (below 4 psu; Bogutskaya et al., 2013).

### 5.4 Conservation implications

Since 1990, the RSL has been part of the Danube Delta Biosphere Reserve, established to preserve the genetic
diversity of the local flora and fauna (Rezervatia Biosferei - Delta Dunarii 2017). Although the protection focuses on
the restriction of human impact on the deltaic system today, no actions have been undertaken to reduce the negative





effects of past human activity. Over the past decades, Pontocaspian species occurrences and abundances have
declined strongly within the RSL system. We are in need of new detailed observation campaigns that specifically
target areas where Pontocaspian species were found in our cores in order to assess whether low numbers of these
species locally may be present. The current study demonstrates that a restoration of previous salinity gradients by
human intervention is the most likely solution to reverse the steep decline in Pontocaspian species richness and
abundance. The development of dynamic barriers that allow limited and managable influx of sea waters, similar as
developed elsewhere (Banning et al., 2018) might be a strategy to be considered. Such an approach would enable
restoration of the salinity gradients and dynamics in the RSL system and also enable mobile oranisms such as fish to
migrate between low salinity areas in the RSL and the mesohaline Black Sea. Such actions would benefit both the
unique benthic and fish populations in the RSL.

Factors that are typically seen as negative influences on natural systems, such as projected global sea-level rise as
well as increased coastal erosion following sediment starvation due to damming of the Danube Delta (Panin and Jipa,
2002), might actually be helpful in the restoration of such conditions as new outlets will be created naturally.

### 5.5 The value of conservation palaeobiological case studies

This study serves as an example of how conservation issues can be targeted by analysing a detailed palaeobiological
record of environmental and faunal change. The Pontocaspian community lives within a salinity gradient and is
constrained between marine and freshwater environments. Its presence can be traced through time and space in the
RSL record. Our approach of using a palaeobiological record that is suitable for outlining direct conservation
consequences might well be applied to other biota under pressure. Mangrove and reef slope communities are good
examples; these can contain high-resolution palaeobiological records (Cramer et al., 2017). However, addressing
post-depositional mixing of faunas using a taphonomic approach is required to assess the quality and fidelity of the
palaeobiological record.

### 6 Conclusions

We present a palaeobiological case study of threatened Pontocaspian biota from the Razim-Sinoe Lake system along
the Romanian Black Sea coast. Our 2000 year record shows the existence of a Pontocaspian community that shifted
through the system along salinity gradients that in turn were influenced by both natural processes and human
interventions. The near complete breakdown of the salinity gradients in the past decades corresponds with a major
decline in these threatened biota. We argue for the restoration of salinity gradients in order to protect the
Pontocaspian habitats and species. Documenting a palaeobiological record can only be successful when
postdepositional taphonomic processes are taken into account.

### Author contribution

FPW, WK, SV, ELJ, CVB designed the study; ELJ, SR, MS, ABP, AMG, SV, FPW, LP conducted fieldwork and
collected the material; SV, ELJ, SR, ABA processed the material; ELJ, CVB, HS realized the age model; ELJ, HAA



performed the sedimentological analysis; SV, FPW, SR, ABP, AMG identified the species; TAN, SV performed the statistical analysis; the manuscript was written by SV, ELJ, TAN and FPW with input from all co-authors. All authors gave final approval for publication.

**Acknowledgements**

We are thankful to Andrei Briceag and Dani Grosu (GeoEcoMar) for assisting in the field, Piet van Gaever (NIOZ, Texel) for supporting Pb dating, Pierre Plard-Taine (UniLasalle, Beauvais) for contributing to sedimentary data acquisition, Marjan Helwerda (Naturalis Biodiversity Center, Leiden) for assisting with mollusc samples sorting, Wim Kuijper (Leiden University) for helping with freshwater mollusc species identification, and Willem Renema

(Naturalis Biodiversity Center, Leiden) for advise on statistical analyses. We thank Marius Skolka (Universitatea Ovidius Constanța) and Oana Popa (National Museum of Natural History "Grigore Antipa") for sharing information on the currently living species. SV, ELJ, FPW are part of the PRIDE ("Pontocaspian RIse and DEmise") project, which has received funding from the European Union's Horizon 2020 research and innovation program under the Marie Sklodowska-Curie grant agreement No 642973. TAN was supported by an Alexander-von-Humboldt

Scholarship.

**Conflict of interest**

The authors declare no conflict of interests.

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



**Table 1 Core data.**

| Expedition | Core | Latitude | Longitude | Lake | Water depth (m) | Core length (m) |
|---|---|---|---|---|---|---|
| 2015 | C1 | 44.538111°N | 28.778583°E | Sinoie | 1.10 | 0.97 |
| 2015 | C2 | 44.665944°N | 28.937139°E | Sinoie | 1.90 | 1.20 |
| 2015 | C3 | 44.694194°N | 28.995028°E | Goloviţa | 0.80 | 1.19 |
| 2015 | C4 | 44.698000°N | 28.937167°E | Goloviţa | 0.50 | 1.10 |
| 2015 | C5 | 44.893472°N | 29.036167°E | Razim | 2.40 | 0.59 |
| 2015 | C6 | 44.872722°N | 28.874194°E | Razim | 1.50 | 1.20 |
| 2016 | C7 | 44.863889°N | 29.096944°E | Razim | 1.30 | 1.16 |
| 2016 | C9 | 44.800000°N | 28.987778°E | Razim | 3.50 | 0.60 |
| 2016 | C10 | 44.789167°N | 28.918056°E | Razim | 1.70 | 1.60 |
| 2016 | C11 | 44.714444°N | 28.786111°E | Goloviţa | 1.50 | 1.92 |
| 2016 | C13 | 44.736667°N | 28.844444°E | Goloviţa | 1.50 | 1.95 |




**Table 2 Results of the *a posteriori* test of the contributions of individual species to the overall concordance of the four groups identified by K-means clustering at *k* = 4. Provided are the mean Spearman correlation between a species and all other species in the respective group, its contribution to the overall concordance statistic *W* for that group, the permutational probability corrected using Holm's method and the respective group it is associated with. Species belonging to the non-concordant group and those not significantly associated (P > 0.05) are marked with an asterisk. Note that, except for the outlier *Rissoa membranacea*, groups 1–3 correlate well with the three defined salinity groups.**


| Species | Spearman mean | *W* per species | Corrected P | Group |
|---|---|---|---|---|
| *Dreissena polymorpha* | 0.187 | 0.35 | 0.032 | 1 |
| *Dreissena bugensis* | 0.226 | 0.381 | 0.03 | 1 |
| *Gyraulus crista** | 0.13 | 0.304 | 0.23 | 1 |
| *Lithoglyphus naticoides** | 0.16 | 0.328 | 0.175 | 1 |
| *Valvata piscinalis* | 0.222 | 0.378 | 0.029 | 1 |
| *Adacna fragilis* | 0.451 | 0.561 | 0.002 | 2 |
| *Clathrocaspia knipowitschii* | 0.435 | 0.548 | 0.002 | 2 |
| *Hypanis plicata* | 0.423 | 0.538 | 0.002 | 2 |
| *Monodacna colorata* | 0.434 | 0.547 | 0.002 | 2 |
| *Rissoa membranacea* | 0.36 | 0.488 | 0.002 | 2 |
| *Abra segmentum* | 0.495 | 0.596 | 0.002 | 3 |
| *Cerastoderma glaucum* | 0.452 | 0.561 | 0.002 | 3 |
| *Ecrobia maritima* | 0.534 | 0.627 | 0.002 | 3 |
| *Mytilaster minimus* | 0.336 | 0.469 | 0.002 | 3 |
| *Parthenina interstincta* | 0.35 | 0.48 | 0.002 | 3 |
| *Lentidium mediterraneum** | -0.064 | 0.149 | 0.83 | 4 |
| *Potamopyrgus antipodarum** | 0.138 | 0.311 | 0.113 | 4 |
| *Retusa truncatula** | 0.08 | 0.264 | 0.299 | 4 |
| *Theodoxus danubialis** | 0.104 | 0.283 | 0.23 | 4 |
| *Theodoxus fluviatilis** | 0.019 | 0.216 | 0.729 | 4 |