# Peer review of "A conservation palaeobiological approach to assess faunal response of threatened biota under natural and anthropogenic environmental change"

_Biogeosciences, 2019_

## Short Comment (SC1) · 13 Feb 2019

This is an interesting manuscript describing a palaeoecological study in part of the Danube Delta. It is, like many papers about conservation palaeobiology, basically a detailed palaeoecological study with a brief mention of conservation palaeobiology at the beginning and the end of the manuscript. It seems to be that the manuscript is more suited to a biological or palaeoecological journal such as Palaeo-3, Quaternary International, or some of the marine, freshwater, or aquatic journals such as Hydrobiologia.

[Figure]

Specific comments:

line 104: fresh water or freshwater – please be consistent

line 132: Did the PVC pipes have a piston? Obtaining a reliable 3m long core with an open PVC tube sounds fraught with problems.

line 136: What was 14C dated – bulk sediment, terrestrial macrofossils?

line 137: How was the calibration done – OxCal, Bchron, Bacon?

lines 193-203: Why did you not use ter Braak's canonical correspondence analysis to give you a direct gradient analysis rather than this rather complex two-stage procedure?

line 204: Given you have 3 a priori groups (your evolutionary species groups), why not do a direct multiple discriminant analysis using the 3 groups? This too can be done in ter Braak's Canoco program with the unique advantage that the statistical significance of your a priori groupings can be tested using permutation tests.

line 205: Standardisation (subtracting the mean and dividing by the standard deviation) has the effect of giving all taxa equal weight. Is that what you want here?

line 210: What are 'the most encompassing assemblages'?

lines 230-233: Hardly worth saying as Deep-time sediments usually experience post-depositional compaction.

line 403: Is there a word missing, as the sentence does not make sense?

line 540, advice, not 'advise'.

―――――――――――――――――

---

## Referee Comment (RC1) · Paolo G. Albano (Referee) · 4 Mar 2019

This paper addresses the effects of natural and human environmental modifications during the last 2000 years in an area in the southern Danube river delta and to its endemic so-called Pontocaspian species. The study highlights how such modifications, and especially the most recent human-induced ones, are causing the local eradication of these species. Given that their distributional area is very restricted, the loss of this habitat and its endemic species is of major conservation concern. By highlighting the historical causes of habitat modification, the study offers practical solutions fitting the

growing need of conservation studies based on baseline data and understanding of the area temporal dynamics and puts paleoecological data into an effective framework for conservation decision making.

I have few comments mostly to invite the Authors to discuss the uncertainties implied by some of the limitations of their approach.

- line 58-59: as it is written, the sentence suggests that conservation paleobiology studies like those cited do not bear a policy relevant conservation message; I assume that this is unintentional, because in my opinion those papers do offer a practical conservation message. Some rewording may be necessary.

- line 130: please specify the core diameters. Please provide more details on how the cores (especially the long ones) were collected.

- line 135: the Authors should be very careful in using the date of first occurrence of a non-indigenous species to date an horizon in a core because time lags in first detection are the norm (Crooks 2005). Due to such time lags, the first occurrence in a core may indicate a time which is considerably before the first published record, as demonstrated for the invasive bivalve Anadara transversa in the Adriatic (Albano et al 2018): the introduction history reconstructed from cores was three times longer than based on the first published occurrence report. This uncertainty must be acknowledged. Moreover, the Authors should write in greater detail the introduction history of Potamopyrgus in the studied region (rather than in Europe in general) providing the year of first report for the site closest to the study area. Crooks JA (2005) Lag times and exotic species: the ecology and management of biological invasions in slow-motion. Ecoscience 12 (3): 316-329. Albano PG et al (2018) Historical ecology of a biological invasion: the interplay of eutrophication and pollution determines time lags in establishment and detection. Biological Invasions 20 (6): 1417-1430.

- The number of 14C dated samples is really small and, according to S5, is limited to one sample per core (generally at the bottom, I assume to constrain the maximum

age). However, there is no reference to time averaging and mixing when interpreting the results of this approach. Please, also specify which species you 14C dated and provide greater detail the calibration procedure.

- line 173: please define "GeoEcoMar"

- line 180-182: apparently there is no reference to the confounding factors such as time-averaging and mixing-bioturbation when interpreting the results. Please note that focusing on specimens poorly affected by taphonomic processes (lines 180-182) does not provide any guarantee against their effects because shells which get quickly buried may display very low taphonomic damage and be mixed in the sediments due to e.g. bioturbation. See also Tomasovych et al (2018) for an example of species co-occurrence in a core which mask differential variation of production in time. Tomasovych et al (2018) A decline in molluscan carbonate production driven by the loss of vegetated habitats encoded in the Holocene sedimentary record of the Gulf of Trieste. Sedimentology 10.1111/sed.12516

- line 354: please specify if the Pontocaspian species you did not encounter alive survive elsewhere.

- Fig. 6: I appreciate this figure which shows the studied organisms.

- Supplements: please list the species in systematic (and not alphabetic) order (e.g. in S2, S7). Note that the file and sheet naming sometimes do not match (e.g. S2 and S3, S7).

---

## Author Comment (AC1) · 5 Apr 2019

**Subject:**
Authors' Response to RC1 (Paolo G. Albano)

**Text:**
Thank you for the comments in the public review process of our paper. The suggestions and the discussion on the uncertainties applied by our approach will improve the manuscript. Please find our responses to the comments below in blue.

This paper addresses the effects of natural and human environmental modifications during the last 2000 years in an area in the southern Danube river delta and to its endemic so-called Pontocaspian species. The study highlights how such modifications, and especially the most recent human-induced ones, are causing the local eradication of these species. Given that their distributional area is very restricted, the loss of this habitat and its endemic species is of major conservation concern. By highlighting the historical causes of habitat modification, the study offers practical solutions fitting the growing need of conservation studies based on baseline data and understanding of the area temporal dynamics and puts paleoecological data into an effective framework for conservation decision making. I have few comments mostly to invite the Authors to discuss the uncertainties implied by some of the limitations of their approach.

- line 58-59: as it is written, the sentence suggests that conservation paleobiology studies like those cited do not bear a policy relevant conservation message; I assume that this is unintentional, because in my opinion those papers do offer a practical conservation message. Some rewording may be necessary.

**Answer**: We checked the sentence carefully again. We will rephrase it to:

'Altogether the data should not only focus on scientific observations and understanding, but also contribute to conservation policies with relevant proposals for ecosystem management (Albano et al., 2016; Helama et al., 2007; Kosnik and Kowalewski, 2016; Martinelli et al., 2017; Vegas-Vilarrubia et al., 2011).'

- line 130: please specify the core diameters. Please provide more details on how the cores (especially the long ones) were collected.

**Answer**: We will give more detailed information and change line 130-131 to:

'We performed facies and fauna analyses on eleven shallow sediment cores. The lakes are very shallow and did not allow boats with piston core facilities. We manually took PVC cores with a diameter of 7.5 cm and lengths between 0.5 m and 1.95 m at water depths of 1.0 to 3.5 m during two expeditions in October 2015 and July 2016 (Fig. 1b; Table 1). By pushing the pipes in the soft sediment and ensuring they were sealed we were able to create a vacuum when retrieving the cores. Deformation along the core edges was minimal. Sedimentary structures like horizontal beddings were conserved within the sediments proving that the sampling method was successful and representative for changes through time and space of the depositional environment.'

- line 135: the Authors should be very careful in using the date of first occurrence of a non-indigenous species to date an horizon in a core because time lags in first detection are the norm (Crooks 2005). Due to such time lags, the first occurrence in a core may indicate a time which is considerably before the first published record, as demonstrated for the invasive bivalve *Anadara transversa* in the Adriatic (Albano et

al 2018): the introduction history reconstructed from cores was three times longer than based on the first published occurrence report. This uncertainty must be acknowledged. Moreover, the Authors should write in greater detail the introduction history of *Potamopyrgus* in the studied region (rather than in Europe in general) providing the year of first report for the site closest to the study area.

Crooks JA (2005) Lag times and exotic species: the ecology and management of biological invasions in slow-motion. Ecoscience 12 (3): 316-329.
Albano PG et al (2018) Historical ecology of a biological invasion: the interplay of eutrophication and pollution determines time lags in establishment and detection. Biological Invasions 20 (6): 1417-1430.

**Answer**: We acknowledge the uncertainty of using a first occurrence date of certain non-indigenous species. We checked again the literature carefully for the first report of *Potamopyrgus* in our region of study, Romania and the Black Sea, which appears to be later than for other parts of Europe: around 1951-1952.

We will therefore add and change information on the introduction history of *Potamopyrgus* after line 154:
'Finally, the first arrival of the invasive snail *Potamopyrgus antipodarum* (Gray, 1843) in Europe provided an additional maximum age tie point. The species is originally from New Zealand, and the chronology of its introduction in other parts of the world is well known. It was introduced around 1859 to England, in 1872 to Tasmania, in 1895 to mainland Australia, in ca. 1900 to the European mainland (Ponder 1988), and in 1987 to North America (Zaranko et al. 1997). The first reports of *P. antipodarum* from sites closest to the RLS come from Romania in 1951 (Son 2008) and the Black Sea in 1952 (Gomoiu et al 2002).'

Gomoiu, M.-T., B. Alexandrov, N. Shadrin and Y. Zaitsev. 2002. p. 341-350. In: E. Leppakoski, S. Gollasch and S. Olenin [eds.]. Invasive Aquatic Species of Europe. Distribution, Impacts and Management. Kluwer Academic Publishers, Boston.
Son, M.O. 2008. Rapid expansion of the New Zealand mud snail Potamopyrgus antipodarum (Gray, 1843) in the Azov-Black Sea Region. Aquatic Invasions, 3(3): 335-340.
Therriault, T. W., Weise, A. M., Gillespie, G. E., Morris, T. J., & Department of Fisheries and Oceans, Ottawa, ON(Canada); Canadian Science Advisory Secretariat, Ottawa, ON(Canada). (2011). Risk assessment for New Zealand mud snail(Potamopyrgus antipodarum) in Canada (No. 2010/108). DFO, Ottawa, ON(Canada).
Zaranko, D. T., Farara, D. G., & Thompson, F. G. (1997). Another exotic mollusc in the laurentian great lakes: the New Zealand native Potamopyrgus antipodarum (Gray 1843)(Gastropoda, Hydrobiidae). Canadian Journal of Fisheries and Aquatic Sciences, 54(4), 809-814.

- The number of 14C dated samples is really small and, according to S5, is limited to one sample per core (generally at the bottom, I assume to constrain the maximum age). However, there is no reference to time averaging and mixing when interpreting the results of this approach. Please, also specify which species you 14C dated and provide greater detail the calibration procedure.

**Answer**: Although the amount of [14]C dated samples is very small, we believe that the combination of different dating methods is strong enough to create our age model ([210]Pb stable isotopes, [14]C dating, magnetostratigraphy, the introduction of *Potamopyrgus*). Indeed the samples were taken at the bottom to constrain the maximum age with C14 and at the top to constrain a younger age with Pb210. Furthermore, the reviewer is right that we should refer to the problem of time averaging (Kidwell, 2002). We will add a sentence after line 232:

'The age control does not allow detailed insights into reworking through bioturbation and time averaging. However, the consistency of assemblages and observed trends indicates in general low rates of reworking. This suggest that the active layer in most of the system has been very shallow.'

For more information on $^{14}$C dated species we will add another table, Table S9:

| Core | Depth (cm) | Species | Amount of specimens | Weight (in mg, min. 20 mg) |
|------|-----------|---------|---------------------|----------------------------|
| C-03 | 114 | *Lentidium mediterraneum* | 6 | 40 |
| C-06 | 120 | *Ecrobia maritima* | 16 | 32 |
| C-07 | 78 | *Monodacna* sp fragments | 8 | 30 |
| C-07 | 137 | *Dreissena polymorpha* | 1 | 31 |
| C-10 | 66 | *Dreissena polymorpha* fragments | 6 | 32 |
| C-13 | 120 | *Lithoglyphus naticoides* | 1 | 21 |
| C-13 | 180 | *Ecrobia maritima* | 15 | 30 |

- line 173: please define "GeoEcoMar"

**Answer**: We will change line 173-174 to:
'Van Veen grab and dredge samples from 77 stations obtained during a 2017 survey of GeoEcoMar (the National Institute for Research and Development of Marine Geology and Geoecology, Romania) were investigated for living molluscs.

- line 180-182: apparently there is no reference to the confounding factors such as time-averaging and mixing-bioturbation when interpreting the results. Please note that focusing on specimens poorly affected by taphonomic processes (lines 180- 182) does not provide any guarantee against their effects because shells which get quickly buried may display very low taphonomic damage and be mixed in the sediments due to e.g. bioturbation. See also Tomasovych et al (2018) for an example of species co-occurrence in a core which mask differential variation of production in time.

Tomasovych et al (2018) A decline in molluscan carbonate production driven by the loss of vegetated habitats encoded in the Holocene sedimentary record of the Gulf of Trieste. Sedimentology 10.1111/sed.12516

**Answer**: Thank you for that comment The reviewer has very valid point here and our approach is in this sense somewhat crude. We will add after line 182:
'We are aware of confounding processes that may result in mixed assemblages with specimens that have a similar preservation signature. Species that burrow into a layer where good preserved molluscs from an earlier period are present may yield the same final good preservation status (Tomasovych et al., 2018). As we cannot correct for that in the analysis it is treated in the discussion on the interpretation of the results.'

We will add in the Discussion after line 475:
'It is also known that taphonomic filters do not always provide a guarantee against the effect of taphonomic processes. Shells can quickly get buried by sediment and therefore show little taphonomic damage, but still get mixed in the sediments afterwards due to bioturbation (Tomasovych 2018). Only large scale dating of shells with very accurate dating methods might solve this issue, yet the existing dating techniques are not accurate enough to address this issue.'

- line 354: please specify if the Pontocaspian species you did not encounter alive survive elsewhere.

**Answer**: We will add in the discussion (after line 494):
'We are uncertain as to the present day occurrence of Pontocaspian species *Clathrocaspia knipowitchi* and *Adacna fragilis*, even though we did found paired bivalves of the latter species in beach material along the Taganrog Bay (Sea of Azov in 2017 suggesting it is alive there. No recent records of living *Hypanis plicata* exist for the Black Sea Basin, but fresh material has been found along Caspian shores (Wesselingh et al., 2019)'

- Fig. 6: I appreciate this figure which shows the studied organisms.

**Answer**: Thank you.

- Supplements: please list the species in systematic (and not alphabetic) order (e.g. in S2, S7). Note that the file and sheet naming sometimes do not match (e.g. S2 and S3, S7).

**Answer**: We will change both supplements accordingly.

---

## Author Comment (AC2) · 5 Apr 2019

**Subject:**
Authors' Response to SC1 (John Birks)

**Text:**
Thank you for carefully reviewing our manuscript and providing a referee comment in the public review process. Your constructive comments will improve our manuscript. Please find our answers below in blue.

This is an interesting manuscript describing a palaeoecological study in part of the Danube Delta. It is, like many papers about conservation palaeobiology, basically a detailed palaeoecological study with a brief mention of conservation palaeobiology at the beginning and the end of the manuscript. It seems to be that the manuscript is more suited to a biological or palaeoecological journal such as Palaeo-3, Quaternary International, or some of the marine, freshwater, or aquatic journals such as Hydrobiologia.

ANSWER: We think that our paper fits the scope of Biogeosciences really well. According to the journal's scope as stated on their webpage, it '…review[s] papers on all aspects of the **interactions** between the biological, chemical, and physical processes. The objective of the journal is to cut across the boundaries of established sciences and achieve an interdisciplinary view of these interactions.' We studied how the interactions between natural and human environmental modification over the past 2000 years effected the endemic mollusc species communities. We use lithological and sedimentological criteria, palaeomagnetic approaches, $Pb^{210}$ datings, $C^{14}$ measurements, mollusc composition and taphonomy. More important, we integrate very recent observation and collection data that underline the urgent need for action, which makes this paper well different from other conservation paleobiological papers. Our paper is a model example of an interdisciplinary study on the boundaries of five fields that are covered by Biogeosciences: biodiversity and ecosystem functioning, biogeochemistry, sedimentary records and palaeobiology.

Specific comments:

line 104: fresh water or freshwater – please be consistent

ANSWER: We actually were not inconsistent but chose the spelling according to the following definition: "Freshwater is an adjective used to describe inland bodies of water and things that live in water that is not salty. It is two words—fresh water—when it doesn't function as an adjective. So a freshwater lake, for instance, is one that has fresh water, and a freshwater fish is one that lives in fresh water." (https://grammarist.com/spelling/freshwater/). We will check the document carefully again for consistency.

line 132: Did the PVC pipes have a piston? Obtaining a reliable 3m long core with an open PVC tube sounds fraught with problems.

The lakes are very shallow and do not allow boats with piston core facilities. By pushing the pipes in the soft sediment and ensuring they were sealed we were able to create a vacuum when retrieving the cores. Deformation along the core edges was minimal. Sedimentary structures like horizontal beddings were conserved within the sediments proving that the sampling method was successful and representative for changes through time and space of the depositional environment.

line 136: What was 14C dated – bulk sediment, terrestrial macrofossils?

ANSWER: For more information on [14]C dated species we will add another supporting information table: Table S9:

| Core | Depth (cm) | Species | Amount of specimens | Weight (in mg, min. 20 mg) |
|------|-----------|---------|---------------------|----------------------------|
| C-03 | 114 | *Lentidium mediterraneum* | 6 | 40 |
| C-06 | 120 | *Ecrobia maritima* | 16 | 32 |
| C-07 | 78 | *Monodacna* sp fragments | 8 | 30 |
| C-07 | 137 | *Dreissena polymorpha* | 1 | 31 |
| C-10 | 66 | *Dreissena polymorpha* fragments | 6 | 32 |
| C-13 | 120 | *Lithoglyphus naticoides* | 1 | 21 |
| C-13 | 180 | *Ecrobia maritima* | 15 | 30 |

line 137: How was the calibration done – OxCal, Bchron, Bacon?

ANSWER:
The [14]C ages have been calibrated to calendar years with the software program: OxCal, version 4.3 (Bronk Ramsey, 2017). We used the calibration curve: IntCal13 (Reimer et al., 2013: IntCal13 and Marine13 radiocarbon age calibration curves 0–50,000 years cal BP, Radiocarbon 55(4):1869–1887). We will add this information in the Material and Methods section.

lines 193-203: Why did you not use ter Braak's canonical correspondence analysis to give you a direct gradient analysis rather than this rather complex two-stage procedure?

ANSWER: We actually considered using CCA at first, but after in-depth study of the literature, we concluded that the analysis is not well suited for our dataset. In their book on Numerical Ecology, Borcard et al. (2011, p. 198) stated the following: "Two important conditions are that the species must have been sampled along their whole ecological range and that they display unimodal responses toward their main ecological constraints.". Neither of both conditions are met in our case: 1) Some of the species occur under different conditions outside the Razim-Sinoie Lake complex (e.g. in the Caspian Sea or Black Sea lagoons), and 2) the ecological constraints of most species occurring there are poorly studied, so we cannot estimate their detailed response. Because of these limitations we chose to perform an nMDS, which is not bound to these assumptions. There, the surface modelling of environmental data (salinity and grain size) does not affect the ordination plot.

line 204: Given you have 3 a priori groups (your evolutionary species groups), why not do a direct multiple discriminant analysis using the 3 groups? This too can be done in ter Braak's Canoco program with the unique advantage that the statistical significance of your a priori groupings can be tested using permutation tests.

ANSWER: Using a discriminant analysis would be indeed a useful approach to test if the three groups defined a priori (based on salinity alone) are significantly different. However, our approach was a slightly different one. We wanted to assess if there are ecological groupings (which might be based on more than just salinity). Showing that those groupings indeed correspond largely to the three groups defined

by salinity indicates the relevance of salinity for the species involved. Such an inference would not be possible using discriminant analysis alone.
Kendall's test also includes an a posteriori test to assess the significance of each group.

line 205: Standardisation (subtracting the mean and dividing by the standard deviation) has the effect of giving all taxa equal weight. Is that what you want here?

ANSWER: Yes, indeed, this kind of data transformation was on purpose. It was suggested by Borcard et al. (2011, p. 79). We will explain it in a bit more detail in the revision to avoid confusion.

line 210: What are 'the most encompassing assemblages'?

ANSWER: By 'the most encompassing assemblages' we mean the assemblages that have the smallest number of clusters with the largest number of positively and significantly associated species (see line 211-212). We are not certain whether the reviewer may have missed this explicit explanation.

lines 230-233: Hardly worth saying as Deep-time sediments usually experience postdepositional compaction.

ANSWER:  the referral to compaction is in our opinion very relevant. It affects depositional rates as reconstructed here. Outside specialized sedimentological experts, the role of compaction is often not appreciated, hence our referral.

line 403: Is there a word missing, as the sentence does not make sense?

ANSWER: Indeed there is mistake in this sentence. It should read: 'Both the presence of marine Association III species in all cores **and the salinity estimations of mesohaline conditions (Fig. 4)** confirm the idea of a marine bay and show the mesohaline waters from the Black Sea dominated most of the RSL.'

line 540, advice, not 'advise'.

ANSWER: We will change it.

---

## Author Response (AR1)

Dear editor,

Thank you for sending us your reply to our paper *"A conservation palaeobiological approach to assess faunal response of threatened biota under natural and anthropogenic environmental change"*. As you proposed, we have revised the manuscript according to the suggestions made during the online reviews. The suggestions will definitely improve the manuscript. Please find our responses to the comments below in blue.

Authors' Response to RC1 (Paolo G. Albano)

This paper addresses the effects of natural and human environmental modifications during the last 2000 years in an area in the southern Danube river delta and to its endemic so-called Pontocaspian species. The study highlights how such modifications, and especially the most recent human-induced ones, are causing the local eradication of these species. Given that their distributional area is very restricted, the loss of this habitat and its endemic species is of major conservation concern. By highlighting the historical causes of habitat modification, the study offers practical solutions fitting the growing need of conservation studies based on baseline data and understanding of the area temporal dynamics and puts paleoecological data into an effective framework for conservation decision making. I have few comments mostly to invite the Authors to discuss the uncertainties implied by some of the limitations of their approach.

- line 58-59: as it is written, the sentence suggests that conservation paleobiology studies like those cited do not bear a policy relevant conservation message; I assume that this is unintentional, because in my opinion those papers do offer a practical conservation message. Some rewording may be necessary.

**Answer**: We checked the sentence carefully again. We rephrased it to:

'Altogether the data should not only focus on scientific observations and understanding, but also contribute to conservation policies with relevant proposals for ecosystem management (Albano et al., 2016; Helama et al., 2007; Kosnik and Kowalewski, 2016; Martinelli et al., 2017; Vegas-Vilarrubia et al., 2011).'

- line 130: please specify the core diameters. Please provide more details on how the cores (especially the long ones) were collected.

**Answer**: We gave more detailed information and changed line 130-131 to:

'We performed facies and fauna analyses on eleven shallow sediment cores. Because the RSL lakes are very shallow they do not allow for boats with piston core facilities. We manually took PVC cores with a diameter of 7.5 cm and lengths between 0.5 m and 1.95 m at water depths of 1.0 to 3.5 m during two expeditions in October 2015 and July 2016 (Fig. 1b; Table 1). By pushing the pipes in the soft sediment and ensuring they were sealed we were able to create a vacuum when retrieving the cores. Deformation along the core edges was minimal. Sedimentary structures like horizontal beddings were conserved within the sediments proving that the sampling method was successful and representative for changes through time and space of the depositional environment. The cores were cut lengthwise in half back in the laboratory.'

- line 135: the Authors should be very careful in using the date of first occurrence of a non-indigenous species to date an horizon in a core because time lags in first detection are the norm (Crooks 2005). Due to such time lags, the first occurrence in a core may indicate a time which is considerably before the first published record, as demonstrated for the invasive bivalve *Anadara transversa* in the Adriatic (Albano et al 2018): the introduction history reconstructed from cores was three times longer than based on the first published occurrence report. This uncertainty must be acknowledged. Moreover, the Authors should write in greater detail the introduction history of *Potamopyrgus* in the studied region (rather than in Europe in general) providing the year of first report for the site closest to the study area.

Crooks JA (2005) Lag times and exotic species: the ecology and management of biological invasions in slow-motion. Ecoscience 12 (3): 316-329.

Albano PG et al (2018) Historical ecology of a biological invasion: the interplay of eutrophication and pollution determines time lags in establishment and detection. Biological Invasions 20 (6): 1417-1430.

**Answer**: We checked again the literature carefully for the first report of *Potamopyrgus* in our region of study (Romania and the Black Sea), which appears to be later (1951-1952) than for other parts of Europe (1859-1900). We also acknowledge the uncertainty of using a first occurrence date of certain non-indigenous species, as indeed it might post-date the date of true arrival. To correct for the possible time lag, we decided to use the earliest date of the first arrival in Europe (1859) and not the first report of arrival in the Black Sea (1952). As the true time of arrival will always be uncertain, we only used the arrival date as an estimation of the maximum age tie point. Afterwards we checked the consistency of the age model by recalculating the model with the Black Sea arrival date to see if a possible time lag would change the consistency of our model and conclusions, and it did not.

We have therefore added and changed the information on the introduction history of *Potamopyrgus* after line 156:

'The species is originally from New Zealand, and the chronology of its introduction in other parts of the world is well known. It was introduced around 1859 to England, in 1872 to Tasmania, in 1895 to mainland Australia, in ca. 1900 to the European mainland (Ponder, 1988), and in 1987 to North America (Zaranko et al, 1997). The first reports of *P. antipodarum* from sites closest to the RLS come from Romania in 1951 (Son, 2008) and the Black Sea in 1952 (Gomoiu et al, 2002). As the first detection of an alien species possibly post-dates the timing of its true arrival (Albano et al, 2018; Crooks, 2005), we use the earliest known arrival date of 1859 in Europe and not the first report of arrival in the Black Sea. We checked the consistency of the age model by recalculating the model with the Black Sea arrival date to see if a possible time lag would severely affect our model and conclusions.'

In the results we added after line 232:

'Moreover, our age model was little affected (1–5 years difference) by the use of the first arrival date of *P. antipodarum* in Europe (1859) compared to the first documented Black Sea region occurrence in 1952.'

- The number of 14C dated samples is really small and, according to S5, is limited to one sample per core (generally at the bottom, I assume to constrain the maximum age). However, there is no reference to time averaging and mixing when interpreting the results of this approach. Please, also specify which species you 14C dated and provide greater detail the calibration procedure.

**Answer**: Although the amount of $^{14}$C dated samples is very small, we believe that the combination of different dating methods is strong enough to create our age model ($^{210}$Pb stable isotopes, $^{14}$C dating, magnetostratigraphy, the introduction of *Potamopyrgus*). Indeed the samples were taken at the bottom to constrain the maximum age with C14 and at the top to constrain a younger age with Pb210. Furthermore, the reviewer is right that we should have referred to the problem of time averaging (Kidwell, 2002). We have added the following sentence after line 232:

'We excluded admixed reworked faunas (high taphonomic scores) to reduce a possible bias in our age model. Any other bias from reworking through bioturbation and time-averaging is very unlikely given the consistency of assemblages and observed trends.'

For more information on $^{14}$C dated species we have added another table, Table S9, providing information on the $^{14}$C dated species.

- line 173: please define "GeoEcoMar"

**Answer**: We have changed line 173-174 to:
'Van Veen grab samples and dredge samples from 77 stations obtained during a 2017 survey of GeoEcoMar (the National Institute for Research and Development of Marine Geology and Geoecology, Romania), were investigated for living molluscs.'

- line 180-182: apparently there is no reference to the confounding factors such as time-averaging and mixing-bioturbation when interpreting the results. Please note that focusing on specimens poorly affected by taphonomic processes (lines 180- 182) does not provide any guarantee against their effects because shells which get quickly buried may display very low taphonomic damage and be mixed in the sediments due to e.g. bioturbation. See also Tomasovych et al (2018) for an example of species co-occurrence in a core which mask differential variation of production in time.

Tomasovych et al (2018) A decline in molluscan carbonate production driven by the loss of vegetated habitats encoded in the Holocene sedimentary record of the Gulf of Trieste. Sedimentology 10.1111/sed.12516

**Answer**: We have added to the Results, after line 232:

'We excluded admixed reworked faunas (high taphonomic scores) to reduce a possible bias in our age model. Any other bias from reworking through bioturbation and time-averaging is very unlikely given the consistency of assemblages and observed trends.'

In the Discussion we changed line 474-475 to:

'This incompatibility may have resulted from the mixing of successive communities by, e.g. bioturbation in a single interval. Species that burrow into a layer containing well preserved molluscs from an earlier period may yield a similar preservation status (Tomašových et al., 2018), even though we tried to avoid such influence by using taphonomic filters.'

- line 354: please specify if the Pontocaspian species you did not encounter alive survive elsewhere.

**Answer**: We have added to the discussion (after line 494):
'Very small numbers of Adacna fragilis and high numbers of living Hypanis plicata have been reported in 2007–2008 in Lake Golovița (Popa et al., 2009). No other Pontocaspian mollusc species have recently been reported alive in the RSL. Outside the RSL, but within the Black Sea Basin, Clathrocaspia knipowitschii has been reported alive in 2005 in the Lower Dnieper area, Ukraine (Anistratenko, 2013).'

- Fig. 6: I appreciate this figure which shows the studied organisms.

- Supplements: please list the species in systematic (and not alphabetic) order (e.g. in S2, S7). Note that the file and sheet naming sometimes do not match (e.g. S2 and S3, S7).

**Answer**: We have changed both the supplements accordingly. We did not notice any mistake in the sheet naming.

Authors' Response to SC1 (John Birks)

This is an interesting manuscript describing a palaeoecological study in part of the Danube Delta. It is, like many papers about conservation palaeobiology, basically a detailed palaeoecological study with a brief mention of conservation palaeobiology at the beginning and the end of the manuscript. It seems to be that the manuscript is more suited to a biological or palaeoecological journal such as Palaeo-3, Quaternary International, or some of the marine, freshwater, or aquatic journals such as Hydrobiologia.

**Answer**: We think that our paper fits the scope of Biogeosciences really well. According to the journal's scope as stated on their webpage, it '…review[s] papers on all aspects of the **interactions** between the biological, chemical, and physical processes. The objective of the journal is to cut across the boundaries of established sciences and achieve an interdisciplinary view of these interactions.' We studied how the interactions between natural and human environmental modification over the past 2000 years effected the endemic mollusc species communities. We use lithological and sedimentological criteria, palaeomagnetic approaches, Pb$^{210}$ datings, C$^{14}$ measurements, mollusc composition and taphonomy. More important, we integrate very recent observation and collection data that underline the urgent need for action, which makes this paper well different from other conservation paleobiological papers. Our paper is a model example of an interdisciplinary study on the boundaries of five fields that are covered

by Biogeosciences: biodiversity and ecosystem functioning, biogeochemistry, sedimentary records and palaeobiology.

170     Specific comments:

line 104: fresh water or freshwater – please be consistent

**Answer**: We actually were not inconsistent but chose the spelling according to the following definition: "Freshwater
175     is an adjective used to describe inland bodies of water and things that live in water that is not salty. It is two words—fresh water—when it doesn't function as an adjective. So a freshwater lake, for instance, is one that has fresh water, and a freshwater fish is one that lives in fresh water." (https://grammarist.com/spelling/freshwater/). We have checked the document carefully again for consistency.

180     line 132: Did the PVC pipes have a piston? Obtaining a reliable 3m long core with an open PVC tube sounds fraught with problems.

**Answer:** We have added a few sentence on the sampling design. After line 130 we have added:

185     'Because the RSL lakes are very shallow they do not allow for boats with piston core facilities. We manually took PVC cores with a diameter of 7.5 cm and lengths between 0.5 m and 1.95 m at water depths of 1.0 to 3.5 m during two expeditions in October 2015 and July 2016 (Fig. 1b; Table 1). By pushing the pipes in the soft sediment and ensuring they were sealed we were able to create a vacuum when retrieving the cores. Deformation along the core edges was minimal. Sedimentary structures like horizontal beddings were conserved within the sediments proving
190     that the sampling method was successful and representative for changes through time and space of the depositional environment.

line 136: What was 14C dated – bulk sediment, terrestrial macrofossils?

195     **Answer**: We have added a supplementary table (S9) providing extra information on the $^{14}$C dated species.

line 137: How was the calibration done – OxCal, Bchron, Bacon?

200     **Answer**: We have added extra information to the Material and Methods section, after line 137:

'The 14C ages have been calibrated to calendar years with the software program OxCal version 4.3 (Ramsey, 2017), using the calibration curve IntCal13 (Reimer et al., 2013). For the Late Holocene, reservoir ages have been estimated at approximately 450 to 550 years in the Danube Delta (Bonsall et al., 2004) and between 250 and 500 years in the
205     Black Sea (Kwiecien et al., 2008). For more information on the 14C dated species see Table S9.'

lines 193-203: Why did you not use ter Braak's canonical correspondence analysis to give you a direct gradient analysis rather than this rather complex two-stage procedure?

210     **Answer**: We actually considered using CCA at first, but after in-depth study of the literature, we concluded that the analysis is not well suited for our dataset. In the book on Numerical Ecology, Borcard et al. (2011, p. 198) stated the following: "Two important conditions are that the species must have been sampled along their whole ecological range and that they display unimodal responses toward their main ecological constraints.". Neither of both conditions are met in our case: 1) Some of the species occur under different conditions outside the Razim-Sinoie Lake complex
215     (e.g. in the Caspian Sea or Black Sea lagoons), and 2) the ecological constraints of most species occurring there are poorly studied, so we cannot estimate their detailed response. Because of these limitations we chose to perform an nMDS, which is not bound to these assumptions. There, the surface modelling of environmental data (salinity and grain size) does not affect the ordination plot.

220     line 204: Given you have 3 a priori groups (your evolutionary species groups), why not do a direct multiple discriminant analysis using the 3 groups? This too can be done in ter Braak's Canoco program with the unique advantage that the statistical significance

of your a priori groupings can be tested using permutation tests.

225 **Answer**: Using a discriminant analysis would be indeed a useful approach to test if the three groups defined a priori (based on salinity alone) are significantly different. However, our approach was a slightly different one. We wanted to assess if there are ecological groupings (which might be based on more than just salinity). Showing that those groupings indeed correspond largely to the three groups defined by salinity indicates the relevance of salinity for the species involved. Such an inference would not be possible using discriminant analysis alone. Kendall's test also
230 includes an a posteriori test to assess the significance of each group.

line 205: Standardisation (subtracting the mean and dividing by the standard deviation) has the effect of giving all taxa equal weight. Is that what you want here?

235 **Answer**: Yes, indeed, this kind of data transformation was on purpose. It was suggested by Borcard et al. (2011, p. 79). We have explained it in a bit more detail to avoid confusion. We changed line 204-205 to:

'The data were Hellinger-transformed, i.e. square-rooting the relative abundances of count data, and standardized by subtracting the mean and dividing by the standard deviation (Borcard et al., 2011; Legendre and Gallagher, 2001).'
240
line 210: What are 'the most encompassing assemblages'?

**Answer**: By 'the most encompassing assemblages' we mean the assemblages that have the smallest number of clusters with the largest number of positively and significantly associated species (see line 211-212). We are not
245 certain whether the reviewer may have missed this explicit explanation.

lines 230-233: Hardly worth saying as Deep-time sediments usually experience postdepositional compaction.

**Answer**: the referral to compaction is in our opinion very relevant. It affects depositional rates as reconstructed here.
250 Outside specialized sedimentological experts, the role of compaction is often not appreciated, hence our referral.

line 403: Is there a word missing, as the sentence does not make sense?

**Answer**: Indeed there is mistake in this sentence. We changed it to: 'Both the presence of marine Association III
255 species in all cores **and the salinity estimations of mesohaline conditions (Fig. 4)** confirm the idea of a marine bay and show the mesohaline waters from the Black Sea dominated most of the RSL.'

line 540, advice, not 'advise'.

260 **Answer**: We have changed it.

We hope that we have dealt with the comments of the reviewers appropriately and that you can agree with our approach. We look forward to hearing from you.

265 Kind regards,

Sabrina van de Velde & Liesbeth Jorissen

[revised manuscript text omitted]